# Mixture of Synthetic Plant Volatiles Attracts More Stick Tea Thrips *Dendrothrips minowai* Priesner (Thysanoptera: Thripidae) and the Application as an Attractant in Tea Plantations

**DOI:** 10.3390/plants13141944

**Published:** 2024-07-15

**Authors:** Zhengwei Xu, Guowei Zhang, Yan Qiu, Zongxiu Luo, Xiaoming Cai, Zhaoqun Li, Lei Bian, Nanxia Fu, Li Zhou, Fida Hussain Magsi, Zongmao Chen, Xiaoming Zhang, Chunli Xiu

**Affiliations:** 1Tea Research Institute, Chinese Academy of Agricultural Sciences, Hangzhou 310008, China; x19912851474@126.com (Z.X.); zhangguowei2003@126.com (G.Z.); luozongxiu@tricaas.com (Z.L.); cxm_d@tricaas.com (X.C.); zqli@tricaas.com (Z.L.); bianlei@tricaas.com (L.B.); funanxia@tricaas.com (N.F.); lizhou@tricaas.com (L.Z.); fidajanmagsi@yahoo.com (F.H.M.); zmchen2006@163.com (Z.C.); 2National Key Laboratory for Conservation and Utilization of Biological Resources in Yunnan, College of Plant Protection, Yunnan Agricultural University, Kunming 650201, China; 3Weifang Natural Resources and Planning Bureau, Weifang 261000, China; wfszmz@163.com; 4Key Laboratory of Biology, Genetics and Breeding of Special Economic Animals and Plants, Ministry of Agriculture and Rural Affairs, Hangzhou 310008, China

**Keywords:** *Dendrothrips minowai*, attractant, behavioral responses, field trapping

## Abstract

The stick tea thrip (*Dendrothrips minowai*) is one of the most serious sucking pests of tea plants (*Camellia sinensis*) in China, North Korea, and Japan. Plant volatile lures are widely used for both monitoring and mass trapping. Previously, we demonstrated that sticky traps baited with *p*-anisaldehyde, eugenol, farnesene, or 3-methyl butanal captured significantly more *D. minowai* in tea plantations, with *p*-anisaldehyde notably capturing the most. In this study, we showed that *D. minowai* adults exhibited significantly higher attraction to mixtures of *p*-anisaldehyde, eugenol, and farnesene compared to an equivalent dose of *p*-anisaldehyde alone in H-tube olfactometer assays under laboratory conditions. Moreover, in field experiments conducted in 2022, rubber septa impregnated with a ternary blend of *p*-anisaldehyde, eugenol, and farnesene (at 3–4.5 mg and a ratio of 3:1:1) captured the highest number of adults on sticky traps, outperforming traps bailed with individual components or a solvent control over two weeks. Significantly, the mass trapping strategy employing these lures achieved control efficacies ranging from 62.8% to 70.7% when compared to traps without attractant, which achieved control efficacies of only 14.2% to 35.4% across three test sites in 2023. These results indicate that the combination of *p*-anisaldehyde, eugenol, and farnesene exhibits an additive or synergistic effect on *D. minowai*. In conclusion, our findings establish a theoretical framework and provide practical technological support for integrating attractant-based strategies into comprehensive thrips management strategies.

## 1. Introduction

The tea industry is a pivotal force in poverty alleviation and rural revitalization within China’s tea-growing region [1,2]. With increasing restrictions on pesticide usage, there is an urgent need for research and development to find efficient and environmentally safe alternatives to tea cultivation [3,4]. Concurrently, there is a growing consumer demand for tea that is safe and healthy and has lower pesticide residues, reflecting a broader trend towards healthier lifestyles [5,6]. However, these perennial woody plants primarily distributed in subtropical and tropical areas with warm climates and ample rainfall create ideal conditions for the proliferation of tea pests [7]. In recent years, the prevalence and severity of small pests such as leafhoppers and thrips, which use their sucking mouthparts to feed on tea plants, have escalated [8,9,10]. Prevention and control of these pests, particularly small ones like thrips, often rely heavily on chemical pesticides due to lacking sustainable green management practices [11,12]. Consequently, there is a critical need to develop safe and efficient green control measures explicitly tailored for small pests in tea plantations. 

Plant volatiles mediate the interactions between plants and pests, influencing pest population dynamics [13,14]. Utilizing plant volatiles to develop attractants and non-toxic pest control techniques has become a research hotspot in plant protection [15,16]. One practical approach is attract-and-kill, which combines plant-derived attractants with trapping devices to lure and eliminate pests [17,18,19]. This method offers advantages such as avoiding direct contact with crops or beneficial organisms; effectiveness against both sexes of pests; and safety for natural enemies, humans, and the environment [18,20,21]. Currently, plant-derived attractants have proven effective in monitoring and controlling various pests, including some species of flies, lepidoptera, thrips, and beetles [22,23,24]. Consequently, optimizing substances and formulations with strong attractant properties remains paramount. 

The stick tea thrip, *Dendrothrips minowai* Priesner (Thysanoptera: Thripidae), is a regional fulminant but easily overlooked sucking pest of tea plants, *Camellia sinensis* (L.) Kuntze (Theaceae), in China, North Korea, and Japan [25,26]. Both nymphs and adults of *D. minowai* feed on fresh tea leaves by sucking sap, while female adults lay their eggs in tender leaves, leading to symptoms such as brown stains, leaf curling, and ultimately reduced tea yields and economic losses [25,27]. Current control measures heavily rely on pesticide applications, which have led to resistance issues, harm to natural enemies, and pesticide residues in tea products [12,26]. Hence, there is an urgent need for green control measures for thrips, with plant-based attractants playing a pivotal role in integrated pest management strategies. 

In previous studies, we identified *p*-anisaldehyde, eugenol, farnesene, and 3-methyl butanal as individual plant volatiles that significantly attract *D. minowai* [28]. However, the potential synergistic effects between these plant volatiles remain unclear. To explore more efficient attractant formulations, this study first assessed the behavioral responses of female and male *D. minowai* to mixtures of these four candidate plant volatiles using an H-tube olfactometer under controlled laboratory conditions. Subsequently, we evaluated whether these volatile mixtures could outperform *p*-anisaldehyde, which previously exhibited the highest trapping efficacy, in field trials. Finally, field bioassays were conducted to confirm the influence of these plant volatiles on the thrips population, as well as to identify effective, long-lasting, plant-derived attractants. This research contributes to advancing commercial traps that utilize plant volatiles as a crucial component of integrated pest management strategies for thrips in tea plantations.

## 2. Results

### 2.1. Behavioral Responses of D. minowai to Odorant Mixtures in the Laboratory

Overall, both female and male *D. minowai* adults showed a marked preference for zones treated with combinations of odorants compared to equal amounts of *p*-anisaldehyde alone, and this preference significantly differed from the combinations of *p*-anisaldehyde and 3-methyl butanal (Figure 1a,b). Among these combinations, the blend of *p*-anisaldehyde, eugenol, and farnesene demonstrated the strongest stimulatory effect. However, there were no significant differences observed between this blend and other combinations (treatments 1 and 3–6) regarding attraction in *D. minowai* at 1 h (females: *F*_1,18_ = 42.88, *p* < 0.001; males: *F*_1,18_ = 81.85, *p* < 0.001). The averages number per repetition was 11.2, 2.60 times that of the single stimulant, indicating obvious synergism upon adding eugenol and farnesene to *p*-anisaldehyde. Furthermore, the attractiveness of the formulation decreased after following the addition of 3-methyl butanal to the mixture (treatments 1, 3, 4, and 7).

### 2.2. Field Trapping of D. minowai Using Combinations of p-Anisaldehyde, Eugenol, and Farnesene

The field experiments demonstrated that traps baited with ternary blends of *p*-anisaldehyde, eugenol, and farnesene significantly outperformed traps baited with single volatiles (18 May 2022: *F*_3,12_ = 31.63, *p* < 0.001; 21 May 2022: *F*_3,12_ = 35.39, *p* < 0.001; 24 May 2022: *F*_3,12_ = 27.04, *p* < 0.001; 27 May 2022: *F*_3,12_ = 22.53, *p* < 0.001) and the control (18 May 2022: *F*_4,15_ = 66.24, *p* < 0.001; 21 May 2022: *F*_4,15_ = 88.09, *p* < 0.001; 24 May 2022: *F*_4,15_ = 59.58, *p* < 0.001; 27 May 2022: *F*_4,15_ = 33.72, *p* < 0.001) in tea plantation (Figure 2). 

### 2.3. Optimal Dosage of Attractant for Mass Trapping

During field trials conducted on 3, 6, and 9 June 2022, the number of *D. minowai* adults trapped by lures containing ternary blends was significantly higher than that of the control (3 June 2022: 69.3 ± 6.89 per trap; 6 June 2022: 74.3 ± 5.36; 9 June 2022: 70.0 ± 10.02 vs. 3 June 2022: *F*_3,8_ = 145.37, *p* < 0.001; 6 June 2022: *F*_3,8_ = 120.40, *p* < 0.001; 9 June 2022: *F*_3,8_ = 19.68, *p* < 0.001) (Figure 3). Moreover, the results unequivocally demonstrated that the number of *D. minowai* adults trapped on sticky traps loaded with the lures containing ternary blends increased with the dosage increasing from 1.5 to 4.5 mg/septum. However, dosage optimization field tests indicated that sticky traps loaded with ternary blends at 3 mg/septum trapped nearly the same number of *D. minowai* adults as those loaded with 4.5 mg/septum (3 June 2022: *F*_1,4_ = 0.15, *p* = 0.359; 6 June 2022: *F*_1,4_ = 0.14, *p* = 0.729; 9 June 2022: *F*_1,4_ = 2.00, *p* = 0.231), indicating that the optimal dosage for mass trapping of thrips was between 3 and 4.5 mg/septum.

### 2.4. The Optimal Attractant (p-Anisaldehyde, Eugenol, and Farnesene) Ratio for Mass Trapping

Within the experimental ratios, the numbers of *D. minowai* adults trapped on sticky traps by lures loaded with different proportions of attractant volatiles were significantly higher than the control (12 June 2022: *F*_4,10_ = 145.37, *p* < 0.001; 15 June 2022: *F*_4,10_ = 120.40, *p* < 0.001; 18 June 2022: *F*_4,10_ = 19.68, *p* < 0.001) (Figure 4). Among these, sticky traps loaded with a ratio of *p*-anisaldehyde, eugenol, and farnesene at 3:1:1 trapped the highest number of *D. minowai* adults, with average counts per trap of 231.0 ± 13.61, 355.7 ± 6.64 and 222.7 ± 8.41, respectively. This ratio resulted in approximately twice the captures compared to other ratios, indicating that the optimal proportion of *p*-anisaldehyde, eugenol, and farnesene for mass trapping of thrips was 3:1:1 per septum. 

### 2.5. Duration of Attractant Efficacy

Based on observed continuous trapping efficacy at four-day intervals, a significantly greater number of thrips were consistently attracted to the ternary attractant compared to the control in tea plantation at different sites during half a month (Shaoxing: 8 October 2022: *F*_1,6_ = 589.55, *p* < 0.001; 12 October 2022: *F*_1,6_ = 333.12, *p* < 0.001; 16 October 2022: *F*_1,6_ = 417.69, *p* < 0.001; 20 October 2022: *F*_1,6_ = 191.33, *p* < 0.001; 24 October 2022: *F*_1,6_ = 16.24, *p* < 0.001; Yuhang: 2 November 2022: *F*_1,6_ = 170.08, *p* < 0.001; 6 November 2022: *F*_1,6_ = 101.84, *p* < 0.001; 10 November 2022: *F*_1,6_ = 81.17, *p* < 0.001; 14 November 2022: *F*_1,6_ = 41.72, *p* < 0.001; 18 November 2022: *F*_1,6_ = 22.04, *p* = 0.003) (Figure 5). The number of thrips trapped on sticky traps decreased day by day; however, there was no significant difference in the number of thrips trapped on traps baited with the attractant up to 20 days after placement at both sites (Shaoxing: 28 October 2022: *F*_1,6_ = 0.35, *p* = 0.574; Yuhang: 22 November 2022: *F*_1,6_ = 1.06, *p* = 0.343). 

### 2.6. Mass Trapping Trials for D. minowai

Initially, there were no significant differences in the number of *D. minowai* adults per 100 leaves among treatment sites within the same tea plantation before the placement of sticky traps (Table 1). However, seven days after deploying traps baited with either solvent or attractant, the numbers of *D. minowai* adults significantly decreased compared to the blank control plots (Shaoxing: traps baited with attractant, 92.6 ± 15.63, *F*_1,8_ = 275.64, *p* < 0.001; traps baited with solvent, 198.4 ± 25.54, *F*_1,8_ = 90.73, *p* < 0.001; Hangzhou: traps baited with attractant, 113.6 ± 10.09, *F*_1,8_ = 84.80, *p* < 0.001; traps baited with solvent, 194.6 ± 54.05, *F*_1,8_ = 10.60, *p* = 0.012; Wenzhou: traps baited with attractant, 70.2 ± 12.62, *F*_1,8_ = 28.81, *p* < 0.001; traps baited with solvent, 140.6 ± 20.21, *F*_1,8_ = 1.254, *p* = 0.295). Similarly, significant decreases in the numbers of *D. minowai* adults were observed 14 days after deploying traps baited with solvent or attractant compared to the blank control. 

Importantly, the control efficiency after 7 and 14 days using the attractant improved significantly by 26.8–56.5% compared to traps alone (i.e., traps baited with solvent) at the same site (7 days: Shaoxing: attractant, 68.7 ± 4.33%, solvent, 28.3 ± 8.16%, *F*_1,8_ = 76.61, *p* < 0.001; Hangzhou: attractant, 57.0 ± 4.78%, solvent, 28.6 ± 8.02%, *F*_1,8_ = 27.95, *p* < 0.001; Wenzhou: attractant, 57.5 ± 6.69%, solvent, 30.7 ± 4.00%, *F*_1,8_ = 47.31, *p* < 0.001; 14 days: Shaoxing: attractant, 70.8 ± 15.73%, solvent, 14.2 ± 7.47%, *F*_1,8_ = 42.20, *p* < 0.001; Hangzhou: attractant, 62.8 ± 13.58%, solvent, 21.7 ± 7.64%, *F*_1,8_ = 27.95, *p* < 0.001; Wenzhou: attractant, 68.7 ± 5.57%, solvent, 35.4 ± 6.70%, *F*_1,8_ = 58.40, *p* < 0.001), demonstrating that mass trapping of *D. minowai* using the attractant had a substantial impact on the thrip adult population.

## 3. Discussion

Utilizing insect behavior responding to chemical signals presents an environmentally friendly approach to pest control [29]. Our previous study demonstrated *p*-anisaldehyde’s efficacy as a lure component for increasing attraction in *D. minowai* [27]. Here, through several experiments, we confirmed that a blend of plant volatiles (*p*-anisaldehyde, eugenol, and farnesene) significantly outperformed solvent controls in attracting *D. minowai*. Furthermore, the synergic effect of *p*-anisaldehyde with the other two substances was approximately 1.3–1.5 times stronger than *p*-anisaldehyde alone (Figure 2). The efficacy of this volatile mixture, including *p*-anisaldehyde, has been validated in both laboratory and field experiments (Figure 1 and Figure 2). Therefore, these semiochemical-baited traps using a blend of plant volatiles (*p*-anisaldehyde, eugenol, and farnesene) offer simple, cost-effective tools widely applicable for integrated pest management (IPM) of thrips. 

Understanding behavioral preferences is crucial for testing and deploying odors effectively in field trapping [30,31]. For various thrips species such as *Frankliniella occidentalis* pergande, *Thrips flavus* Schrank, *Megalurothrips distalis* Karny, and others, their directional responses to plant volatiles such as methyl isonicotinate, benzaldehyde, and nonanal in olfactometers indicate that these compounds serve as alternative attractants [32,33,34]. Therefore, our study initially examined *D. minowai*’s behavioral response to a blend of plant volatiles via H-tube olfactometry in the laboratory to formulate a highly effective attractant. Notably, literature reports on additive or synergistic effects from compound mixtures for thrips are scare, possibly due to the relatively straightforward olfactory mechanisms involved in odorant recognition and transduction [35,36]. Fortunately, we found that female and male *D. minowai* were significantly more attracted to the mixture of *p*-anisaldehyde, eugenol, and farnesene compared to *p*-anisaldehyde alone (Figure 1), which was previously identified as the most attractive compound in our research. Our findings align with a study showing a modest yet significant additive trapping effect for onion *Thrips tabaci* Lindeman using a combination of eugenol and ethyl isonicotinate [37]. 

Increasing the attractant dosage generally correlates with higher pest capture in traps [35,38,39]. Our study assessed the efficacy of blue sticky traps loaded with attractant lures at dosages ranging from 1.5 to 4.5 mg/septum. Our results showed that the number of *D. minowai* adults captured trapped increased proportionally with higher dosages (Figure 3). Optimal attraction occurred between 3 and 4.5 mg/septum, capturing 4.3 to 5.4 times more thrips in a tea plantation compared to the solvent control. Similarly, El-Sayed et al., (2014) found that increasing the dose of ethyl nicotinate from 10 to 500 mg resulted in higher captures of *Thrips obscuratus* (Crawford), nearly 8 to 10 times more thrips captured with 100 and 500 mg compared to 10 mg. Additionally, the specific ratio of volatiles can influence the attractant’s trapping efficacy [40,41]. In our study, the most effective ratio comprised *p*-anisaldehyde, eugenol, and farnesene at 3:1:1 per septum (Figure 4). Similarly, a ternary mixture of ethyl benzoate, 3-carene, and ethyl butyrate (0.05 mg, 1:0.2:5, *v*/*v*/*v*) attracted the highest number of female *Bactrocera dorsalis* (Hendel) in the field trails [42]. 

Plant-derived attractants are proven effective for mass trapping various pest species [43,44]. Our results demonstrated that the ternary attractant captured 1.6 to 11.7 times and 1.8 to 5.4 times more thrips in October and November, respectively, in tea plantations compared to the solvent control (Figure 5). Importantly, our study highlights the field control efficacy of mass trapping using plant-derived attractants to reduce *D. minowai* adult populations. The control efficacy improved by 26.8–56.5% compared to traps with solvent controls across three tested sites (Table 1). This method is particularly effective when thrip populations are low, such as in May. Thus, attractant-enhanced trapping effectively reduces thrip population and presents a promising approach for integrated pest management. Our ongoing evaluations will assess changes in thrips populations throughout the tea plant growing season following widespread attractant application at a large scale.

## 4. Materials and Methods

### 4.1. Thrips 

Tea seedlings (cultivar: Longjing 43) were grown in rectangular plastic pots at the Tea Research Institute, Chinese Academy of Agricultural Sciences in Hangzhou, Zhejiang Province, China (120.10° E, 30.19° N). *D. minowai* adults were collected from the tea plantations of Shaoxing Royal Tea Village Co., Ltd. in Shaoxing, China (120.71° E, 29.94° N) and then reared on the above cultivated tea seedlings in a climate-controlled chamber set at 24 ± 1 °C with a 16 h light/8 h dark cycle and 60 ± 5% relative humidity. Adults of the second generation were used for behavioral experiments.

### 4.2. Reagents

Four compounds, namely, *p*-anisaldehyde, eugenol, farnesene (mixture of isomers, α-farnesene, and (E)-β-farnesene), and 3-methyl butanal, which attracted *D. minowai* in our previous trials, were used as candidates for mixtures [31]. All were purchased from Sigma-Aldrich (St. Louis, MO, USA) (for details, refer to Appendix A). Mineral oil (Sigma-Aldrich, St. Louis, MO, USA) was chosen as solvent in behavioral tests, and hexane [HPLC grade, CNW Technologies GmbH (Düsseldorf, Germany)] was chosen as solvent in field trials. All the above-mentioned plant volatiles were diluted to a specific concentration. 

### 4.3. Olfactometer Bioassays

Behavioral responses of *D. minowai* females and males for different combinations of volatiles discovered previously were analyzed by the modified glass H-tube olfactometers during mid-October and mid-December in a laboratory [31]. The parameters of the H-tube olfactometer that consisted of two straight tubes and one transverse tube are described in the above previous papers. Before formal tests, the H-tube olfactometers were washed by ethanol and then dried for about 1 h at 90 °C in an electric constant-temperature drying oven (Shanghai Sumsung Laboratory Instrument Co., Ltd., Shanghai, China). To prevent thrips from escaping, both ends of the straight tube were sealed with gauze (200 mesh), and transverse tubes were covered with the matching stopper after the thrips were released. 

The individual formula for combining different volatiles containing *p*-anisaldehyde was introduced into one straight tube as olfactory cues, and the equal amount of *p*-anisaldehyde used as the control. A mixture of one, two, three, or four odorants was added on one side of the H-tube, while keeping the other side the same amount of *p*-anisaldehyde. Different formulas for combining different volatiles containing *p*-anisaldehyde comprised (1) mixtures of *p*-anisaldehyde, eugenol, farnesene, and 3-methyl butanal; (2) mixtures of *p*-anisaldehyde, eugenol, and farnesene; (3) *p*-anisaldehyde, eugenol, and 3-methyl butanal; (4) *p*-anisaldehyde, farnesene, and 3-methyl butanal; (5) *p*-anisaldehyde and eugenol; (6) *p*-anisaldehyde and farnesene; and (7) *p*-anisaldehyde and 3-methyl butanal. 

Before testing, 5-day-old unmated thrip adults were starved for about 1 h. For each test, 20 starved thrips were transferred to the middle of the transverse tube with a soft brush and then acclimated for 10 min. After 1 h, the tubes were examined to determine if the thrips crossed the midpoint of the left or the right arm. A choice was considered to have been made if female or male *D. minowai* crossed the midpoint of the left or the right tube and stayed there for more than 30 s; otherwise, it was defined as unresponsive. The total number of thrips that chose the left tube, the right tube, or neither tube was recorded, correspondingly. Each *D. minowai* was used only once and replaced every repetition. At the same time, the H-tube olfactometer was replaced with a clean one after 60 thrips had been tested. Moreover, the treatment direction was reversed for every group. For each combination, there were 10 replicates, and a total of 200 *D. minowai* were evaluated per sex. The tests were conducted in a 50 × 50 × 30 cm chamber covered in a blackout cloth set at 25 ± 1 °C and 60 ± 5% RH. All the bioassays were conducted between 8:00 and 18:00, which is the natural activity of *D. minowai* in tea plantations [45].

### 4.4. Field Tests

A candidate for the attractant formulation for *D. minowai* was determined by the above behavioral tests. The selection was based on the above results showing the mixtures of *p*-anisaldehyde, eugenol, and farnesene to be more attractive to *D. minowai*. The traps consisted of blue sticky plates made of poly (vinyl chloride) (10 cm × 25 cm; Hangzhou Yihao Agricultural Technology Co., Ltd., Hangzhou, China), suspended on poles ≈10 cm above tea plants. The lures consisted of rubber septa (Zhangzhou Ingeer Agricultural Science and Technology Co., Ltd., Zhangzhou, China) impregnated with 3 mg single synthetic plant volatile, or the mixtures dissolved in hexane (used in field experiments 1, 3, 4, and 5). In addition, rubber septa loaded with 100 μL of hexane were used as the control. In all field experiments, blue sticky traps (*n* = 4) baited with lures containing attractant formulation or hexane were placed 6 m apart in a randomized complete block design, and plots were spaced 20 m apart.

To investigate whether the better field attractiveness of combinations of *p*-anisaldehyde, eugenol, and farnesene were consistent with the strong attraction effect in the laboratory, trapping experiments were conducted to examine the average individuals on sticky traps baited with lures containing the ternary blends and every single volatile. In field experiment 1, we examined if the mixtures had a synergistic role in the attraction of thrips and if more thrips were attracted to odors of mixtures. Field trials were conducted to evaluate the trapping and control efficiency of the attractant formulation from the 18 to the 27 May 2022 in tea plantations at the Royal Tea Village Co., Ltd. (120.71° E, 29.94° N) (tea cultivar: Longjing 43), Shaoxing City, China. The lures were loaded with a single plant volatile, the mixtures, or hexane (solvent control). Traps were replaced 3 days after installation, but the lures were not changed and the trapped thrips were counted. 

In field experiments 2 and 3, the trapping efficiency of the attractant lures loaded with different dosages and different proportions were evaluated in tea plantations at the Royal Tea Village Co., Ltd., from the 3 to the 9 June and from the 12 to the 18 June in 2022. Traps and lures were replaced 3 days after installation and the trapped thrips were counted. 

In field experiment 4, the duration of the attractant was evaluated in tea plantations at the Royal Tea Village Co., Ltd., from the 8 to the 28 October in 2022 and in tea plantations at Zhejiang Camel Jiuyu Organic Food Co., Ltd. (119.90° E, 30.40° N) (tea cultivar: Jiukeng), Hangzhou City, China, from the 2 to the 22 November in 2022. Traps were replaced 4 days after installation, but the lures were not changed and the trapped thrips were counted. 

To determine the roles of attractant in the suppression of thrip adult density, we carried out the mass trapping trials by placing sticky traps baited with lures of attractant on large-plot at 3 different sites. In field experiment 5, the controlling efficiency of the attractant was conducted in tea plantations at the Royal Tea Village Co., Ltd., from the 23 May to the 5 June in 2023; in tea plantations at Zhejiang Camel Jiuyu Organic Food Co., Ltd., from the 1 to the 14 June in 2023; and in tea plantations from the 12 to the 25 October in 2023 at Zhejiang Tailong Tea Co., Ltd. (119.57° E, 27.20° N) (tea cultivar: Taishun native population), Wenzhou City, China. The area at each site was about 666 m^2^, blue sticky traps baited with attractant or hexane were placed about 5–6 m apart, and a total of 25 traps were placed for each tea plantation. Before sticky traps were positioned, the population of thrips on tea plants was investigated. Then, the population investigating occurred at intervals of 7 and 14 days following the setting up of the sticky traps. The control efficiency (%) was recalculated as the following formula: 

The decrease rate (%) was recalculated as the following formula: $$\mathrm{Decrease}\;\mathrm{rate}(\%)=\frac{\mathrm{Number}\;\mathrm{of}\;\mathrm{thrips}\;\mathrm{before}\;\mathrm{placement}-\mathrm{Number}\;\mathrm{of}\;\mathrm{thrips}\;\mathrm{after}\;\mathrm{placement}}{\mathrm{Number}\;\mathrm{of}\;\mathrm{thrips}\;\mathrm{before}\;\mathrm{placement}}\times 100$$

### 4.5. Statistical Analysis

For the H-tube olfactometer bioassays, the *t*-test was performed to determine the significance of the difference in the number of thrips between the different treatments (*p* < 0.05). Unresponsive individuals were not included in the analysis, but they were listed at the end of the data bars. For the number of thrips trapped and the thrips population we surveyed in the field experiments, normality (proc univariate) and heteroscedasticity (hovtest in “means” option, proc. glm) were tested before the analysis. Datasets that did not fit the assumption of homogeneity of variance for the analysis were log-transformed [log10 (x + 1)]. Means compared by a one-way ANOVA followed by Duncan’s multiple range test (*p* < 0.05). Graphs were drawn using GraphPad Prism 7.0 and OriginPro 2021. The SAS 9.1 program was used for statistical analyses [46].

## 5. Conclusions

Different combinations, dosages, ratios, trapping efficacy, and control effectiveness of the attractant were evaluated to develop mass trapping technology for controlling *D. minowai*. Our results demonstrate that a combination of *p*-anisaldehyde, eugenol, and farnesene is more attractive to *D. minowai* compared to *p*-anisaldehyde alone or other combinations tested indoors and outdoors. Sticky traps baited with 3–4.5 mg ternary blend impregnated into rubber septa in a ratio of 3:1:1 capture the highest number of thrips in the field. Importantly, deploying traps baited with this attractant significantly reduced *D. minowai* population. Thus, this attractant formulation shows promise for thrip monitoring and mass trapping in tea plantations, particularly effective during the early stage when thrips densities are relatively low.

## Figures and Tables

**Figure 1 plants-13-01944-f001:**
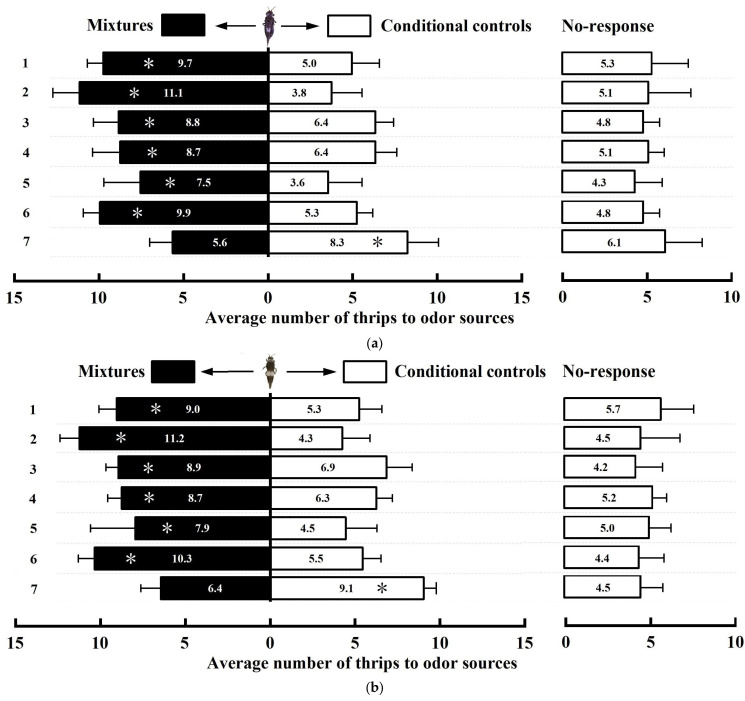
Behavioral responses of *Dendrothrips minowai* unmated females (**a**) and males (**b**) to odorants emitted by mixtures of candidate plant volatiles and the equal amount of *p*-anisaldehyde. 1: mixture of *p*-anisaldehyde, eugenol, farnesene, and 3-methyl butanal vs. the equal amount of *p*-anisaldehyde; 2: mixture of *p*-anisaldehyde, eugenol, and farnesene vs. the equal amount of *p*-anisaldehyde; 3: *p*-anisaldehyde, eugenol, and 3-methyl butanal vs. the equal amount of *p*-anisaldehyde; 4: *p*-anisaldehyde, farnesene, and 3-methyl butanal vs. the equal amount of *p*-anisaldehyde; 5: *p*-anisaldehyde and eugenol vs. the equal amount of *p*-anisaldehyde; 6: *p*-anisaldehyde and farnesene vs. the equal amount of *p*-anisaldehyde; 7: *p*-anisaldehyde and 3-methyl butanal vs. the equal amount of *p*-anisaldehyde. Asterisks indicate a significant difference according to the *t*-test (*p* < 0.05).

**Figure 2 plants-13-01944-f002:**
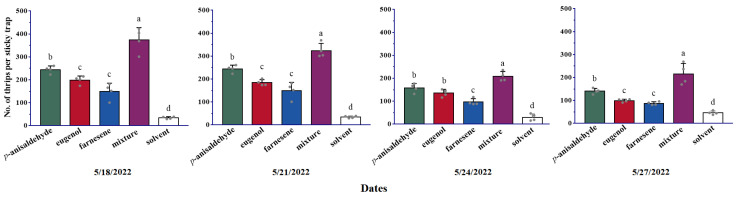
Number of *Dendrothrips minowai* (mean ± SE) captured by sticky traps with lures loaded with single synthetic volatiles, mixtures of synthetic volatiles, and the solvent (control) in a tea plantation at Shaoxing from the 18 to the 27 May 2022. Means with different letters are significantly different (*p* < 0.05) and the same letters are not significantly different (*p* > 0.05) according to a one-way ANOVA followed by Duncan’s multiple range test.

**Figure 3 plants-13-01944-f003:**
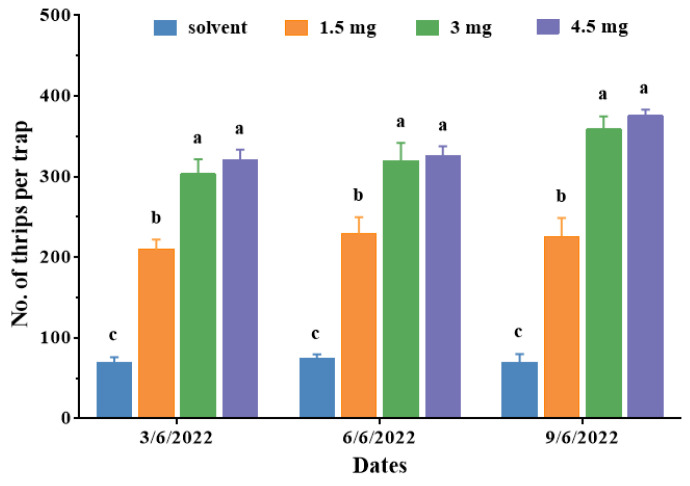
Number of *Dendrothrips minowai* (mean ± SE) captured by sticky traps with lures loaded with different dosages of attractant mixtures in a tea plantation at Shaoxing from the 3 to 9 June 2022. Means with different letters are significantly different (*p* < 0.05) and the same letters are not significantly different (*p* > 0.05) according to a one-way ANOVA followed by Duncan’s multiple range test.

**Figure 4 plants-13-01944-f004:**
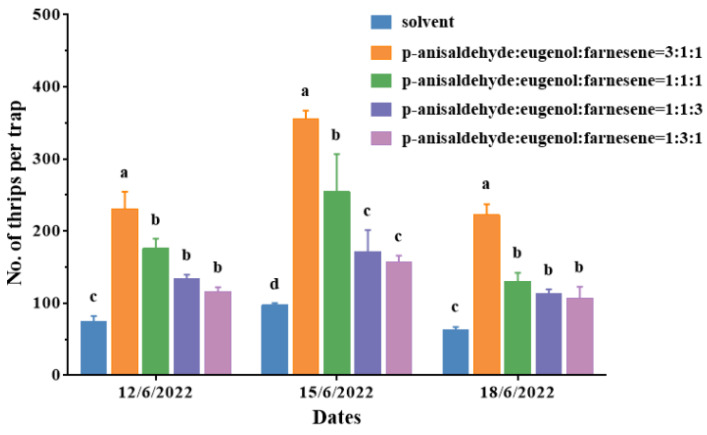
Number of *Dendrothrips minowai* (mean ± SE) captured by sticky traps with lures loaded with different proportions of attractant volatiles in a tea plantation at Shaoxing from the 12 to 18 June in 2022. Means with different letters are significantly different (*p* < 0.05) and the same letters are not significantly different (*p* > 0.05) according to a one-way ANOVA followed by Duncan’s multiple range test.

**Figure 5 plants-13-01944-f005:**
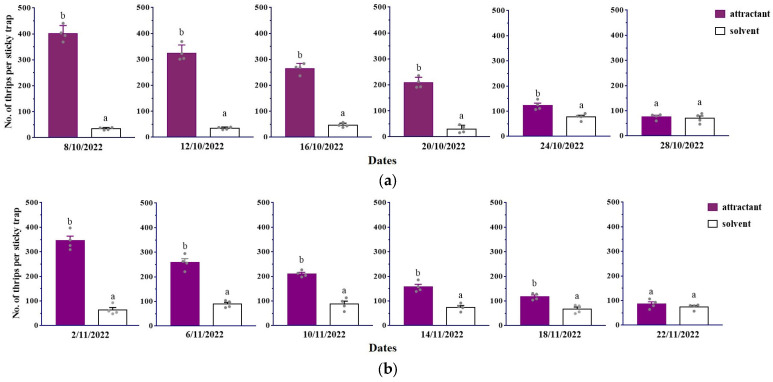
The duration of *Dendrothrips minowai* attractant was evaluated in tea plantations at Shaoxing (**a**) and Hangzhou (**b**) from the 8 to 23 October and from the 2 to 17 November in 2022. Means with different letters are significantly different (*p* < 0.05) and the same letters are not significantly different (*p* > 0.05) according to a one-way ANOVA followed by Duncan’s multiple range test.

**Table 1 plants-13-01944-t001:** The controlling efficiency of the attractant for *Dendrothrips minowai* population determined using sticky traps with lures containing attractant and the solvent control in tea plantations at Shaoxing from the 23 May to the 5 June in 2023, at Hangzhou from the 1 to the 14 June in 2023, and at Wenzhou from the 12 to the 25 October in 2023. Means with different letters are significantly different (*p* < 0.05) and the same letters are not significantly different (*p* > 0.05) according to a one-way ANOVA followed by Duncan’s multiple range test.

Location	Treatments	No. of Thrips before Traps Placed	7 Days after Traps Placed			15 Days after Traps Placed		
			No. of Thrips	Reduced Percent (%)	Control Effect (%)	No. of Thrips	Reduced Percent (%)	Control Effect (%)
Shaoxing	Traps baited with lures of attractant	148.8 ± 29.03 a	92.6 ± 15.63 a	37.0 ± 8.72 c	68.7 ± 4.33 b	36.6 ± 18.97 a	75.6 ± 13.14 b	70.7 ± 15.73 b
	Traps baited with lures of solvent	140.8 ± 34.34 a	198.4 ± 25.54 b	−44.3 ± 16.42 b	28.3 ± 8.16 a	99.4 ± 16.69 b	28.3 ± 6.24 a	14.2 ± 7.47 a
	Blank control	164.0 ± 28.14 a	322.6 ± 26.67 c	−101.2 ± 28.64 a		132.8 ± 15.80 c	16.4 ± 17.94 a	
Hangzhou	Traps baited with lures of attractant	140.8 ± 38.63 a	113.6 ± 10.09 a	16.4 ± 14.55 c	57.0 ± 7.48 b	54.4 ± 13.46 a	59.4 ± 14.83 b	62.8 ± 13.58 b
	Traps baited with lures of solvent	139.2 ± 31.96 a	194.6 ± 54.05 b	−38.9 ± 15.60 b	28.6 ± 8.02 a	119.8 ± 30.63 b	14.5 ± 8.34 a	21.7 ± 7.64 a
	Blank control	163.2 ± 28.6 a	304.2 ± 40.15 c	−94.6 ± 51.04 a		171.0 ± 30.22 c	−9.2 ± 30.09 a	
Wenzhou	Traps baited with lures of attractant	149.0 ± 26.96 a	70.2 ± 12.62 a	52.2 ± 7.55 c	57.5 ± 6.69 b	54.8 ± 13.29 a	63.3 ± 6.54 c	68.7 ± 5.57 b
	Traps baited with lures of solvent	179.2 ± 23.95 b	140.6 ± 25.21 b	21.9 ± 4.51 b	30.7 ± 4.00 a	135.6 ± 25.67 b	24.3 ± 7.86 b	35.4 ± 6.70 a
	Blank control	146.4 ± 8.26 a	163.6 ± 32.43 c	−12.7 ± 24.80 a		170.8 ± 32.63 c	−17.32 ± 24.86 a	

## Data Availability

Data is contained within the article.

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
