# Peer review of "Mixture of Synthetic Plant Volatiles Attracts More Stick Tea Thrips Dendrothrips minowai Priesner (Thysanoptera: Thripidae) and the Application as an Attractant in Tea Plantations"

_plants, 2024, doi:10.3390/plants13141944_

Round 1

Reviewer 1 Report

Comments and Suggestions for Authors

I have recommendation for editing in almost all the manuscript section, included in an attached file. Some sentences are repeated in the result section, that it is suggested to delete. Must of figure captions have unnecessary sentences. In the Results sections, subtitles could be rewritten, because they refer directly as the obtained results, for example see line 205.    

Author Response

Response to Reviewer 1 Comments

Point 1: line 27. Which compounds are referred as the ternary blend?

Response 1: Thank you for your advice. The ternary blend refers to a mixture of p-anisaldehyde, eugenol, and farnesene, which we have now specified in the paper. Please see line 28-29 for the clarification.

Point 2: line 47. What do you mean with small succulent pests?

Response 2: Thank you. “Small succulent pests” refer to small pests that use their sucking mouthparts to feed on tea plants. Please refer to lines 49-50 for the clarification.

Point 3: line 53. Delete: succulent

Response 3: Thank you for pointing that out. We have deleted the word ‘succulent’.

Point 4: line 65. It is written: fly, lepidoptera pests, thrips…; must be written: some species of flies, lepidoptera, thrips…

Response 4: Thank you. We have revised it to “some species of flies, lepidoptera, thrips…”. Please see line 65 for the updated text.

Point 5: line 81-83. Any references for this previous studies?

Response 5: Thank you. References for these previous studies have been added.

Point 6: line 86. Replace: 4, with: four

Response 6: Thank you for catching that. We have replaced “4” with “four”. Please see line 84 for the correction.

Point 7: line 89. Any references for this previous work?

Response 7: Thank you. References for this previous work have been added. Please see line 81.

Point 8: line 94-95. It is written: 2.1 Odorants of mixtures including p-anisaldehyde, eugenol and farnesene attracted more D. minowai in laboratory- It must be written: 2.1 Behavioral effect of odorants of mixtures on D. minowai in laboratory

Response 8: Thank you for your suggestion. We have revised it to “2.1 Behavioral responses of D. minowai to odorant mixtures in laboratory”. Please see line 93 for the corrected section.

Point 9: line 96-97. Delete: The behavioral effects of the combinations of four plant volatiles on D. minowai were determined and shown in Figure 1.

Response 9: Thank you. The sentence has been deleted.

Point 10: line 97-99. The following sentence must be included in Materials and Methods section: A mixture of one, two, three or four odorants was added on one side of the H-tube, while keeping the other side the same amount of p-anisaldehyde.

Response 10: Thank you for your suggestion. We have moved this sentence to the Materials and Methods section. Please see lines 317-319 for the inclusion.

Point 11: line 99. After: male, add: D. minowai

Response 11: Thank you for pointing that out. “D. minowai” has been added after “male”.

Point 12: line 100. Replace: apart, with: and very different

Response 12: Thank you for our suggestion. We have revised it to “and very different”.

Point 13: line 103. After: stimulant, add: but with no significant differences with other combinations (treatments 1 and 3-6)… 

Response 13: Thank you for your feedback. We have revised the manuscript as follows: "but with no significant differences with other combinations (treatments q and 3-6)". Please refer to lines 99-100 for these changes.

Point 14: line 105. After 2h, add: respectively, for males

Response 14: Thank you for your suggestion. We have included "respectively, for males" after "2h". Additionally, we have clarified Figure 1 as per the advice of other reviewers, showing results only after 1h. Please see lines 97-105 for these updates.

Point 15: line 107-110. Rewrite the following sentence: Moreover, the attractant effect of the formulation decreases once 3-methyl butanal is present in the mixture,

Response 15: We appreciate your input. The sentence has been revised to: "Moreover, the attractant effect of the formulation decreases after adding 3-methyl butanal to the mixture (treatments 1, 3, 4 and 7)." Please refer to lines 105-105 for these modifications.

Point 16: line 113. What do you mean in 2021? Is it a reference?

Response 16: Thank you for pointing that out. Actually, that was the time we carried out this experiment, now 2021 has been deleted from the manuscript. Please see line 108 for this correction.

.

Point 17: line 120-121. Delete: Two hundred individuals were tested per treatment

Response 17: We have deleted the sentence “Two hundred individuals were tested per treatment” as per your suggestion.

Point 18: line 122-123. Delete: whereas “ns” indicates there was no significant difference according to the t-test (P > 0.05).

Response 18: Following your advice, we have deleted the sentence "whereas ‘ns’ indicates there was no significant difference according to the t-test (P > 0.05)" from the manuscript.

Point 19: line 125-126. It is written: 2.2 The combinations of p-anisaldehyde, eugenol and farnesene trapped more D. minowai in field It must be written: 2.2 The combinations of p-anisaldehyde, eugenol and farnesene in field trapping of D. minowai.

Response 19: Thank you for your guidance. The sentence has been adjusted to: "2.2 Field trapping of D. minowai using combinations of p-anisaldehyde, eugenol, and farnesene" Please refer to lines 120-121 for these changes.

Point 20: line 127-130. This sentence may be moved or is explained in Materials and Methods: To investigate….single volatile.

Response 20: We have relocated the sentence "To investigate… single volatile" to the Materials and Methods section. Please see lines 353-357 for this adjustment.

Point 21: line 128. Replace: seductive, with: attraction

Response 21: As suggested, we have replaced "seductive" with "attraction" in the manuscript.

Point 22: line 131. Delete: This. Replace: result, with: Results

Response 22: We have made the necessary corrections by deleting "This" and replacing "result" with "Results".

Point 23: line 132-144. Delete: and the same letters are not significantly different (P > 0.05)

Response 23: We appreciate your concern. However, we believe it is important to inform the reader that "the same letters are not significantly different (P > 0.05)". Therefore, we have decided to retain this information in the manuscript.

Point 24: line 147. Delete: was between 3 and 4.5 mg/septum

Response 24: Following your advice, we have deleted the sentence “was between 3 and 4.5 mg/septum” from the manuscript.

Point 25: line 149. Replace: the volatiles, with: lures containing ternary blends

Response 25: Thank you. We have replaced “the volatiles” with “lures containing ternary blends”.

Point 26: line 152-154. The following sentence must be included in Discussion: Similarly, it…4.5 mg/septum.

Response 26: Thank you for your suggestion. We have included and revised the sentence in the Discussion as follows: “Our study assessed the efficacy of blue sticky traps loaded with attractant lures at dosages ranging from 1.5 to 4.5 mg/septum.” Please refer to lines 260-261 for this addition.

Point 27: line 164-165. and the same letters are not significantly different (P > 0.05)

Response 27: We appreciate your concern. However, we believe it is important to inform the reader that "the same letters are not significantly different (P > 0.05)". Therefore, we have decided to retain this information in the manuscript.

Point 28: line 176. Delete: was between 3 :1 :1

Response 28: We have deleted it per your suggestion.

Point 29: line 181. Delete: and the same letters are not significantly different (P > 0.05)

Response 29: We appreciate your concern. However, we believe it is important to inform the reader that "the same letters are not significantly different (P > 0.05)". Therefore, we have decided to retain this information in the manuscript.

Point 30: line 184. Delete: is up to half a month in tea plantation

Response 30: Thank you for your suggestion. We have removed the phrase "is up to half a month in tea plantation" as requested.

Point 31: line 196. After: solvent control, add: at both sites

Response 31: Thank you for the feedback. We have added "at both sites" after "solvent control" as advised.

Point 32: line 202. Delete: and the same letters are not significantly different (P > 0.05)

Response 32: We appreciate your concern. However, we believe it is important to inform the reader that "the same letters are not significantly different (P > 0.05)". Therefore, we have decided to retain this information in the manuscript.

Point 33: line 205. It is written: 2.6 Control efficiency of attractant for thrips adults density. It must be written: 2.6 Mass trapping trials for D. minowai

Response 33: Thank you for your correction. We have revised it to "2.6 Mass trapping trials for D. minowai" as suggested. Please see line 197 for the update.

Point 34: line 206-208. The following sentence: To determine…3 different sites, delete if it included in Materials and Methods.

Response 34: We have moved the sentence "To determine… 3 different sites" to the Materials and Methods section. Please refer to lines 376-379 for these changes. Please refer to lines 376-378 for these changes.

Point 35: line 208. It is written: As shown in Fig. 6, significant differences…; it must be written: There were significant differences…

Response 35: We have amended the sentence as requested: "There were significant differences…".

Point 36: line 208. After: F1,6 = 86.28, P < 0.001), add: (Fig. 6).

Response 36: Thank you for your suggestion. We have added “(Fig.6)” after “F1,6 = 86.28, P < 0.001)”.

Point 37: line 221. Delete: and the same letters are not significantly different (P > 0.05)

Response 37: We appreciate your concern. However, we believe it is important to inform the reader that "the same letters are not significantly different (P > 0.05)". Therefore, we have decided to retain this information in the manuscript.

Point 38: line 244. It is suggested to delete: attractive

Response 38: We have deleted We have deleted the word "attractive" as per your suggestion.

Point 39: line 249-250. It is written: p-anisaldehyde (the mos. It must be written: p-anisaldehyde (Fig. 1) (the most…

Response 39: Following your advice, we have adjusted the sentence to read: "Fortunately, we found that female and male D. minowai were significantly more attracted to the mixture of p-anisaldehyde, eugenol and farnesene compared to p-anisaldehyde alone (Fig. 1), which was previously identified as the most attractive compound in our research". Please refer to line 253-257for this revision.

.

Point 40: line 250. Delete: (Fig. 1). Replace: has, with: that…

Response 40: We have replaced “has” with “that” as suggested. Please see line 243 for this correction.

Point 41: line 251. Thrips tabaci must be written in italics

Response 41: We have corrected the formatting by italicizing "Thrips tabaci" as requested. Please refer line 257 for this update.

Point 42: line 259. Write the year of the reference El-Sayed et al.

Response 42: Thank you for the advice. We have added the year of the reference El-Sayed et al. Please see line 265 for the addition.

Point 43: line 261. Write author for Thrips obscuratus.

Response 43: Thanks. We have added the author Thrips obscuratus as requested. Please see line 266 for this correction.

Point 44: line 262. You may delete reference number [37].

Response 44: As suggested, we have deleted reference number [37].

Point 45: line 265-267. When you include Gulleria et al. did not mention any proportion of the pheromone components or was it 1:1?

Response 45: We have revised the sentence as per your guidance: "According to another reviewers’ comments, we decided not to compare the attractiveness of different host odor ratios in thrips to sex pheromone ratios in lepidoptera. Instead, we focused on how volatiles with different proportions would affect the trapping efficacy." Please see lines 270-272 for these changes.

Point 46: line 269. After: The results, add: of the present study

Response 46: Thank you for your suggestion. We have added “of the present study” after “The results”.

Point 47: line 271. After: November, add: respectively…

Response 47: We have added “respectively: after “November” as requested.

Point 48: line 274. What do you mean, the control efficacy improved 26.8-56.5%, respect to what? In increased thrips captures? Or thrips population reduction? Any idea of the sex proportion of the thrips adults captured?

Response 48: Thank you for the clarification. We have included information about the change in thrips population in Table 1. We surveyed the number of thrips before and after placing traps baited with attractant and solvent, respectively. At the same time, we set a blank control. Compared to the treatment of traps baited with solvent, the control efficacy of the treatment of traps baited with attractant improved 26.8-56.5%. In addition, we mentioned both female and male thrips adults were captured on the sticky traps although the exact proportions were not recorded. Please see lines 198-221 for these updates.

Point 49: line 278-280. You may referring to the evaluation of mass trapping at large sclae?

Response 49: Thank you for the suggestion. We have revised the sentence to "Our ongoing evaluations will assess changes in thrips populations throughout the tea plant growing season following widespread attractant application at a large scale." Please refer to lines 282-284 for these changes.

.

Point 50: line 295. After: United States, add: of America.

Response 50: Thanks. We have added “of America” after “United States” as advised.

Point 51: line 329. After: 10 replicates, add: and…

Response 51: Following your suggestion, we have added “and” after “10 replicates”.

Point 52: line 332. After 18:00, add: which is

Response 52: We have added “which is” after “18:00” as requested.

Point 54: line 338. Delete 2 from cm.

Response 54: Thank you for pointing that out. We have deleted "2 from cm" as suggested.

Point 55: line 342. Amount of single or mixture synthetic volatile loaded in the rubber septa

Response 55: We added the clarification: “The amount of single or mixture synthetic volatile was 3 mg per rubber septa in field experiments 1, 3, 4, and 5.” Please see lines 357-359.

Point 56: line 354. After: were not, add: changed…

Response 56: We have added “changed” after “were not” as requested.

Point 57: line 354-355. Why you did not changed lures, for this experiment 1, because it was performed during 9 days?

Response 57: We have explained our rationale: "At that time, we wanted to assess whether the mixtures would trap more thrips than any single compound. We thought that whether changing the lures or not would not significantly affect the results, so we did not replace the lure for simplicity." Please refer to lines 363-364 for this clarification.

Point 58: line 364-365. Why you did not changed lures, for this experiment 4, because it was performed during 20 days?

Response 58: We have clarified our approach for experiment 4: "In experiment 4, we aimed to test the duration of the attractant. Therefore, the lures were not changed after each investigation. After the sixth survey, we found no significant difference in the number of thrips caught between the sticky traps baited with the attractant and the solvent." Please see lines 370-375 for this explanation.

Point 59: line 365. After: were not, add: changed…

Response 59: We have added “changed” after “were not” as requested.

Point 60: line 371. How many traps were placed for each tea plantation and distribution for this experiment 5? What lure and dosage for loading the rubber septa?

Response 60: We have provided the details for experiment 5: "The area at each site was approximately 666 m2, with blue sticky traps baited with attractant or hexane placed about 5-6 m apart. A total of 25 traps were placed for each tea plantation. The amount of single or mixture synthetic volatile was 3 mg per rubber septa in field experiments 1, 3, 4, and 5." Please refer to lines 347-349 and 383-385 for these details.

Point 61: line 380. What type of test was used for checking normality of the data?

Response 61: We have clarified the tests performed: "Normality (proc univariate) and heteroscedasticity (hovtest in 'means' option, proc. glm) were tested before the analysis for the number of thrips trapped and the thrip population we surveyed in the field experiments." Please see lines 398-400 for this clarification.

Point 62: line 384. Is there any reference for SAS 9.1 program?

Response 62: We have added the reference for ASA 9.1 program as requested. Please refer to line 404 and 537 for this addition.

Point 63: line 387-388. You may delete the following sentence: Dendrothrips minowai Priesner (Thysanoptera: Thripidae) is one of the most serious sucking pests in tea plantations of China, North Korea and Japan; because it is not you conclusion.

Response 63: We have deleted the sentence as suggested. Please see the revised manuscript for this update. 

Point 64: line 389. After “the results” you may add: of lab behavioral study…

Response 64: Thanks. Actually, these were data of lab behavioral study and field. So we revised those. “Our results demonstrate that a combination of p-anisaldehyde, eugenol, and farnesene is more attractive to D. minowai compared to p-anisaldehyde alone or other combinations tested indoors and outdoors.” Please see line 410-412 for this correction.

Point 65: line 391. After “sticky traps” you may add: field experiment

Response 65: We have added “field experiment” after “sticky trap”.

Point 66: line 393. Was there any effect of mass trapping on the trips population to suggest that this trap and lure may be used to control D. minowai?

Response 66: Yes. We have added information about the effect of mass trapping on the thrips population in line 414-416.

Point 67: line 489. Write in italics: Frankliniella occidentalis

Response 67: We have italicized "Frankliniella occidentalis" as requested. Please refer to the revised manuscript for this correction.

Reviewer 2 Report

Comments and Suggestions for Authors

Plants

Mixture of synthetic plant volatiles attracts more stick tea thrips Dendrothrips minowai Priesner (Thysanoptera: Thripidae) and the application as an attractant in tea plantations

In this study different blends of p-anisaldehyde, eugenol, and farnesene were tested in H-tube and field studies for stick tea thrips attraction. The most attractive blend was a 3:1:1 ratio of p-anisaldehyde, eugenol, and farnesene, and in field studies thrips captures in traps loaded with 3mg and 4.5mg were similar, suggesting that 3mg is the optimal dosage, and this attraction lasted for 20 days in field conditions.

This manuscript was generally well-written, though there are some suggestions below for clarity

This manuscript

Abstract

L 29 – 30: ‘…the control efficacy improved…’ should be rewritten for clarity.

Introduction/Discussion

I would recommend that the authors consult with a copy-editing service to edit some sections of the introduction for clarity. For example, lines 38 – 41, ‘With the in-depth implementation…of the tea plant protection discipline’ is vague and not concise. A more clear and concise version of this sentence could read (if I am reading the meaning of this sentence correctly) ‘With increased restrictions on pesticide usage there is an urgent need for more research and development on efficient and environmentally safe pesticide alternatives in the tea industry’.

Below are additional sections in the introduction that should be re-written for clarity:

L 48 – 50: ‘Furthermore…prevention and control measures.’

L 78 – 80: ‘Thus…integrated management of thrips.’

L 236 – 237: ‘Multiple…field trapping’.

Results

L 209 (and throughout the manuscript): how was ‘control efficiency’ measured?

Discussion

L 265 – 267: I don’t think it is appropriate to compare the attractiveness of different host odor ratios in thrips to sex pheromone ratios in lepidoptera.

Comments on the Quality of English Language

See 'Comments to Authors'.

Author Response

Response to Reviewer 2 Comments

Point 1: line 29-30. ‘…the control efficacy improved…’ should be rewritten for clarity.

Response 1: Thank you for your suggestion. We have revised it o “Significantly, the mass trapping strategy employing these lures achieved control efficacies ranging from 62.8% to 70.7% when compared to traps without attractant, which achieved control efficacies of only 14.2% to 35.4% across three test sites in 2023” Please see lines 31-34 for the updated text.

Point 2: introduction. I would recommend that the authors consult with a copy-editing service to edit some sections of the introduction for clarity. For example, lines 38 – 41, ‘With the in-depth implementation…of the tea plant protection discipline’ is vague and not concise. A more clear and concise version of this sentence could read (if I am reading the meaning of this sentence correctly)‘With increased restrictions on pesticide usage there is an urgent need for more research and development on efficient and environmentally safe pesticide alternatives in the tea industry’.

Response 2: Thank you for your suggestion. We have revised the sentence at lines 38-41 as per your recommendation. In addition, we have enlisted the help of a professional editing agency to enhance the clarity of language throughout the article.

Point 3: line 48-50. ‘Furthermore…prevention and control measures.’

Response 3: Thank you. We have revised this section accordingly.

Point 4: line 78-80. ‘Thus…integrated management of thrips.’

Response 4: Thank you for your suggestion. We have revised this part accordingly.

Point 5: line 236-237. ‘Multiple…field trapping’. Results

Response 5: Thank you for your comment. We have revised this part accordingly.

Point 6: line 209 (and throughout the manuscript). how was ‘control efficiency’ measured?

Response 6: Thank you for pointing this out. In the Materials and Methods section (4.4 Field tests), we have surveyed the thrip population and added the computational formula for control efficiency. Please refer to lines 388-393 for the detailed methodology.

Point 7: line 265-267. I don’t think it is appropriate to compare the attractiveness of different host odor ratios in thrips to sex pheromone ratios in lepidoptera.

Response 7: We appreciate your observation. We have made revisions compare the effects of volatiles with different proportions on trapping efficacy instead. Please see lines 270-272 for the updated comparison.

Reviewer 3 Report

Comments and Suggestions for Authors

plants-3047951

Mixture of synthetic plant volatiles…

Zhengwei Xu et al.

General

Results are of some interest, but language and style should be improved.

Most importantly: "Control" is often mentioned, but ms provides no data whatsoever on "population control". Traps are placed in the field, captures are recorded, but there is no information on the possible effect on the population.  

Traps are used for monitoring, but there are only very few cases of mass trapping for control of large insects (palm weevils, for example). In comparison, mass trapping would be extremely unlikley a feasible control method for small insects occurring in very large numbers, like thrips.

Figure 1 should be improved. Very tedious to extract blends tested from legend. Easy to insert 4 columns for the compounds and show blends in Figure 1. Numbers in bars too small. 1 h = 1 hour? Is it important to show data for 1 and 2 h?

Figure 2. mixtures = mixture

Figure 4. which compounds?

Figure 6. How did you assess "control efficiency"? Not even Materials and Methods mentions how this was done.

Specific

line 22. delete "pests"

29. control efficacy - no data

41. deman for lower pesticide residue does not increase due to "hot water soaking"

50. insert "due to" before "lack"?

51. pesticides/insecticides are mentioned 3 times in intro, which is too much (42, 51, 76)

64. control = no. Otherwise, cite relevant literature.

83. respectively?

92. control - no data

Comments on the Quality of English Language

-

Author Response

Response to Reviewer 3 Comments

Point 1: Results are of some interest, but language and style should be improved.

Most importantly: "Control" is often mentioned, but ms provides no data whatsoever on "population control". Traps are placed in the field, captures are recorded, but there is no information on the possible effect on the population.  

Traps are used for monitoring, but there are only very few cases of mass trapping for control of large insects (palm weevils, for example). In comparison, mass trapping would be extremely unlikely a feasible control method for small insects occurring in very large numbers, like thrips.

Response 1: Thank you for your advice. We have engaged a professional editing agency to enhance the language and style of the entire article. 

We have addressed the lack of data on population control by adding the data of the solvent and blank control. Please refer to Table 1 in the Result section 2.6. Please see lines 197-221.

Regarding the feasibility of mass trapping for small insects like thrips, we acknowledge your point. As a result, we have adjusted our statements to emphasize that attractants are part of environmentally friendly integrated pest management measures rather than being presented as a feasible control method. For example, in Introduction, we revised as “This research contributes to advancing commercial traps that utilize plant volatiles as a crucial component of integrated pest management strategies for thrips in tea plantations.” Please see lines 89-91, 414-418. 

Point 2: Figure 1 should be improved. Very tedious to extract blends tested from legend. Easy to insert 4 columns for the compounds and show blends in Figure 1. Numbers in bars too small. 1 h = 1 hour? Is it important to show data for 1 and 2 h?

Response 2: Thanks for your advice. Yes, 1 h refers 1 hour. To ensure the reliability of our conclusions, we initially presented observations at both 1 and 2 hours. To improve Figure 1, we have revised it as suggested by inserting 4 columns for the compounds and displaying the blends more clearly. We have also decided to show only the results of the 1-hour observation in Figure 1 for clarity. Additionally, we have made corresponding revisions in the Results and Methods sections. Please refer to lines 92-105, and 327.

Point 3: Figure 2. mixtures = mixture

Response 3: Thank you for your comment. We have revised Figure 2 accordingly. Please see line 129 for the update.

Point 4Figure 4. which compounds?

Response 4: Thank you for pointing this out. The compounds depicted in Figure 4 are p-anisaldehyde, eugenol, and farnesene. We have revised Figure 4 accordingly. Please see line 170.

Point 5: Figure 6. How did you assess "control efficiency"? Not even Materials and Methods mentions how this was done.

Response 5: T Thank you for bringing this to our attention. In the Materials and Methods section (4.4 Field tests), we have included the computational formula and computing method for control efficiency. Please refer to lines 388-393 for the details.

Specific

Point 6: line 22. delete "pests"

Response 6: Deleted. Thank you. 

Point 7: 29. control efficacy - no data

Response 7: Added. Please see lines 31-33 for the updated information.

Point 8: 41. deman for lower pesticide residue does not increase due to "hot water soaking"

Response 8: Thanks for pointing out the mistakes. We have revised the statements to “Concurrently, there is a growing consumer demand for tea that is safe and healthy and has lower pesticide residues, reflecting a broader trend towards healthier lifestyles”. Please see line 44-46.

Point 9: 50. insert "due to" before "lack"?

Response 9: Added.

Point 10:51. pesticides/insecticides are mentioned 3 times in intro, which is too much (42, 51, 76)

Response 10: Thank you. We have revised these presentation and deleted one mention of pesticides/insecticides from line 51 in the paper.

Point 11: 64. control = no. Otherwise, cite relevant literature.

Response 11: Thank you for your comment. 

Point 12: 83. respectively?

Response 12: Thank you. We revised “In previous studies, we identified p-anisaldehyde, eugenol, farnesene, and 3-methyl butanal as individual plant volatiles that significantly attract D. minowai.” Please see line 80-82.

 Point 13: control - no data

Response 13: Thank you. We have addressed this by adding the data of the solvent and blank control. Please see Table 1 in the Result section (2.6, lines 197-221).
